# Hyperspectral Microscopy Technology to Detect Syrups Adulteration of Endemic Guindo Santo and Quillay Honey Using Machine-Learning Tools

**DOI:** 10.3390/foods11233868

**Published:** 2022-11-30

**Authors:** Guillermo Machuca, Juan Staforelli, Mauricio Rondanelli-Reyes, Rene Garces, Braulio Contreras-Trigo, Jorge Tapia, Ignacio Sanhueza, Anselmo Jara, Iván Lamas, Jose Max Troncoso, Pablo Coelho

**Affiliations:** 1Facultad de Ingeniería, Arquitectura y Diseño, Universidad San Sebastián, Concepción 4080871, Chile; 2Departamento de Física, Universidad de Concepción, Casilla 160-C, Concepción 3349001, Chile; 3Laboratorio de Palinología y Ecología Vegetal, Departamento de Ciencia y Tecnología Vegetal, Universidad de Concepción, Campus Los Angeles, Juan Antonio Coloma 0201, Los Angeles 4451032, Chile; 4Facultad de Ciencias de la Naturaleza, Universidad San Sebastián, Concepción 4080871, Chile; 5School of Electrical Engineering, Pontificia Universidad Católica de Valparaíso, Valparaíso 2362804, Chile; 6Research Center in Natural and Exact Sciences, School of Education and Social Sciences, Adventist University of Chile, Chillan 3780000, Chile

**Keywords:** guindo santo honey, quillay honey, *Eucryphia glutinosa*, *Quillaja saponaria*, hyperspectral microscopy, machine learning, honey adulteration

## Abstract

Honey adulteration is a common practice that affects food quality and sale prices, and certifying the origin of the honey using non-destructive methods is critical. Guindo Santo and Quillay are fundamental for the honey production of Biobío and the Ñuble region in Chile. Furthermore, Guindo Santo only exists in this area of the world. Therefore, certifying honey of this species is crucial for beekeeper communities—mostly natives—to give them advantages and competitiveness in the global market. To solve this necessity, we present a system for detecting adulterated endemic honey that combines different artificial intelligence networks with a confocal optical microscope and a tunable optical filter for hyperspectral data acquisition. Honey samples artificially adulterated with syrups at concentrations undetectable to the naked eye were used for validating different artificial intelligence models. Comparing Linear discriminant analysis (LDA), Support vector machine (SVM), and Neural Network (NN), we reach the best average accuracy value with SVM of 93% for all classes in both kinds of honey. We hope these results will be the starting point of a method for honey certification in Chile in an automated way and with high precision.

## 1. Introduction

The global problem of food fraud is becoming a threat for the entire industry chain, from producers to consumers. Food fraud refers to the intentional substitution, dilution, and aggregation of ingredients in aliments [1]. Food sources often use restoratives, preservatives, and neutralizers as adulterants. It is also a fraudulent practice to mix with lower-quality products, to add water to increase volume, or to use sweetening syrups in dairy products, virgin oils, wines, fruit juices, and honey, among others [2]. Consequently, several industrial food products contain mislabeling and false statements [3].

Regarding regulations and food defense, ISO 22000:2018 became the international standard, focusing on regulating safe food, products, and services that meet customer and regulatory requirements [4]. The implementation of this standard offers multiple methodological and technological challenges for the identification of adulterated foods.

*Apis mellifera* bees, a Hymenoptera order species, make honey, a natural product that is prized for its health advantages and as a sweetener since 6000 BC. According to Trademap data, the average world exportation of honey in 2021 was around 765,000 tons, with a global export value of USD 2650 million [5]. Honey is mostly composed of various sugars (75–85%), including fructose (33–42%) and glucose (27–45%), as well as other compounds obtained from nectar collecting, such as organic acids, minerals, amino acids, enzymes, and solid particles [6,7]. Different bee colonies can have very different diets, and the honey that they produce is determined by the nectar of the flowers, its concentration, and the amount they consume. As a result, distinguishing between sugars that occur naturally and those that have been added later is particularly difficult.

Intentional adulteration in honey is produced primarily through the fraudulent addition of sugar solutions, as these solutions are much cheaper than honey itself [8]. Adulteration also alters the nutritional value of honey, resulting in the loss of most of its properties and health benefits, as well as an economic loss for the beekeeping sector due to the significant discredit towards the product as a consequence of noticeably declining quality and flavor. This fact encourages a decrease in consumption.

In terms of honey adulteration detection, Carbon Isotope Mass Spectrophotometry (SCIRA) corresponds to the official method declared by the AOAC [9], which is why it has been used in several studies [10,11,12,13,14,15]. It is a procedure that evaluates the proportion of carbon isotopes ${}^{13}$C/${}^{12}$C in honey to detect any adulteration, since the ${}^{13}$C isotope is characteristic of C3-type carbohydrates (naturally abundant in honey) and ${}^{12}$C of the carbohydrates C4 (carbohydrate produced from corn and artificially added cane sugar).

SCIRA detects the addition of 7 to 10% exogenous components, but it is expensive and requires highly qualified personnel for its application, making it unsuitable for monitoring, and it cannot detect the addition of syrups produced by C3-type plants such as beet, rice, and wheat [16,17]. Other methodologies have been reported that use liquid or gas chromatography coupled to mass spectrometry [18,19], liquid chromatography with potentiometric detection [20] or with refractive index detection [21], and infrared spectrophotometry [22]. However, all these methods necessitate specialized and costly equipment and highly qualified personnel for their development and subsequent interpretation of results, making the whole process time-consuming.

In this work, we implement a non-targeted method [23,24,25] to detect the purity of Quillay (*Quillaja saponaria*) and Guindo Santo (*Eucryphia glutinosa*), two representative monofloral endemic honeys from Chile, according to their botanical Classification of Chilean honey [26]. For this method, we use hyperspectral transmission microscopy and Köhler illumination to acquire the hyperspectral cubes. From the hyperspectral cubes, spectral signatures are obtained, preprocessed, and dimensionally reduced to classify different levels of purity of honey using supervised machine learning tools with 10-fold cross-validation to protect the overfitting [27]. Note that specific guidelines about selecting the minimal number of samples for training depend on the input problem [28]; in this case, we use 100 curves for each class. Finally, some metrics commonly used in classification problems are calculated to evaluate the method’s performance.

Both exclusive honeys, Guindo Santo and Quillay, suffer from syrup addition fraud, which increases the uncertainty of their purity and limits their commercial growth. With this work, local beekeepers would be incentivized to participate in new markets employing quality seals. Thus, on the one hand, Guindo Santo honey would have an immediate competitive advantage because it is a unique product from the Andes mountain (Alto Biobío in Chile), where the Pewenche indigenous communities live and produce this honey [29]. Finally, on the other hand, Quillay is an endemic species found in the Biobío and Ñuble region, and its honey would also be favored.

## 2. Materials and Methods

Samples of pure syrups of glucose, fructose, maltose, pure honey from Guindo Santo and Quillay, and honey mixed with these syrups, were prepared by means of experimental protocols. Then, the spectral signatures of each were obtained. The information was used to determine whether the honey was adulterated with additional sugars or was pure by using classification algorithms based on artificial learning. The combination of spectral signatures and artificial intelligence reduce the costs of a honey fraud detection equipment and analysis. Once the spectral signatures are obtained, and the artificial intelligence training has been carried out, analysis by specific personnel is no longer necessary.

### 2.1. Honey Sample Collection and Melissopalynological Analysis

The two honey sample collection sites are located in the Biobío region, in Chile, as shown in Figure 1. The first site is a territory with varied endemic vegetation and native flora belonging to the Pewenche community of the Pitril sector (S 37°47′13.7″, W 71°32′11.9″), commune of Alto Biobío, where the Guindo Santo honey production apiaries are encountered. The second site, belonging to Colbún Company, is an area of approximately 125 ha of native species (among them the Quillay endemic tree), inserted in an anthropic matrix of forest plantations of fast-growing species and multiple crops in the commune of Yumbel (S 37°2′2.10″; W 72°40′20.28″). In this place is Quillay’s honey production apiaries. Additionally, it is important to mention that, at each study site, beekeepers placed stamped frames without honey or prior preparation to ensure that when the species under study began producing nectar, the bees produced only Guindo Santo and Quillay honey.

The honey samples were extracted by mechanical centrifugation once the phenological period of each species had finished. For Guindo Santo, the harvest covers the period from 22 January 2020, to 4 February 2020, and for Quillay, the harvest period covers from 15 December 2019, to 20 January 2020. In both seasons, the bees were not receiving supplementary energy or protein feeding. After harvest, the honey samples were analyzed at the Palynology and Plant Ecology Laboratory of the Universidad de Concepción, Los Angeles Campus. Pollen content was determined with the conventional technique of acetolysis according to the method of Faegri and Iversen (1989) [30]. The preparations were deposited in the laboratory Palynotheca for their pollen determination.

Pollen grains were determined at various taxonomic ranges using specialized bibliographic assistance [31,32,33] (Heusser 1971) and the palynological reference collection of the Palynology and Plant Ecology Laboratory of the University of Concepcion, where the Chilean Norm NCh2981 was used [29]. According to this norm, monofloral honey is defined as honey with a percentage equal to or greater than 45%, and bifloral honey is when the honey contains two species in a minimum proportion of 50% and where there is no difference greater than 5% between the two (INN 2007a). Finally, lower percentages correspond to polyfloral content.

The melissopalynological analysis applied to the honey collected from the Pitril sector recorded 15 pollen types for a total of 4 replications counted. The analyzed honey resulted in monofloral origin of the Guindo Santo species with 52.5%. Likewise, for honey collected from Yumbel, the results indicated that the honey analyzed is of monofloral origin from the *Quillaja saponaria* species with 47.7%. It is important to note that there are no significant differences and variability between the counted replicates.

All the honey under study meet the basic national and international standards necessary for their consumption according to the physio-chemical criteria established by the Chilean Standards Division of the National Normalization Institute (INN) for the classification of honey [29].

### 2.2. Honey Adulteration Process

The sample preparation methodology associated with the study of honey by spectral analysis is based on Refs. [11,21,34]. At the beginning of the procedure, in transparent plastic containers with screw caps, different types of honey (Quillay (Q) and Guindo santo (GS)) are mixed with the three different syrups: fructose (F), glucose (G), and high maltose (M), separately, as shown in Figure 2. The mixing proportions used are from 90 to 60% by mass, in such a way that the sum between syrup and honey is 23 g (g), as shown in Table 1. Then, each mixing ratio is stirred manually with a disposable plastic spoon at 20–25 °C until the naked eye detects the homogeneous mixture. Later, reference samples are prepared (syrups and honey samples to 100%, respectively). Finally, each prepared sample is kept at 22 °C, in the absence of sunlight, closed with the container’s lid and parafilm around it to prevent evaporation and contamination until analysis by hyperspectral microscopy imaging. Each trial is performed in duplicate for each honey. We use a single concave microscope slide from Fisher Scientific, ensuring that all samples are identical in their spatial distribution conditions at the moment of measurements.

### 2.3. Experimental Setup

The experimental setup is based on an inverted microscope (Nikon Eclipse Ti-U) equipped with a white light lamp and an objective lens of 20× magnification. The camera is a CMOS, Kiralux CS135MU from Thorlabs, with an array composed of 1200×1920 pixels, coupled with liquid crystal tunable bandpass filters Kurious-VB1 and digital controller from Thorlabs, providing a continuously tunable center wavelength from 420 to 730 [nm]. Narrow, medium, or wide bandwidth can be settled with a Full Width at Half Maximum (FWHM) of 10, 18, and 32 [nm], respectively. The system is presented in Figure 3.

### 2.4. Imaging Procedure

The collection of hyperspectral cubes involves two steps: (i) illumination and (ii) acquisition. In the illumination stage, the idea was to achieve spatially uniform optical intensity (irradiance) at the sample plane, obtaining optimal image quality. Köhler illumination method [35] was implemented to modify the aperture diaphragm and condenser to produce a homogeneous wide field, resulting in a bright image without artifacts and glare. Once the lighting is corrected, hyperspectral data was acquired. For the acquisition step, the prepared samples are positioned on the microscopy slides, and by transillumination, a spectral sweep is performed using the tunable optical bandpass filter digital. The images were acquired from the centers of the cavities containing the samples, and dark room condition was used to avoid external illumination on the sample during measurements.

### 2.5. Data Processing

Spectral curves were obtained from the acquired hyperspectral cubes composed of images for different visible wavelengths. Hyperspectral technology allows us to obtain a spectral curve for each pixel: this means that we have as many radiometric curves as the image has pixels. The total transmittance of the images was calculated by selecting a region of interest (ROI). The ROI is then subdivided into 4 × 4 neighborhoods, which are averaged to create spectral curves. Finally, we measured the transmittance (τ) according to the model $\tau^{\lambda }=P^{\lambda }/W^{\lambda }$[36], where λ is the representation of wavelengths, the incident radiation generated by the light source is $W^{\lambda }$, and the maximum transmitted radiation curves obtained from the neighborhoods’ average is $P_{\lambda }$.

### 2.6. Adulterated Honey Detection Algorithm

To correlate spectra features of the adulterated honey samples and the response of the classification algorithms, it was assumed that the measured spectra were deterministic signals in noise described by an observation model $\rho^{\lambda }=\tau^{\lambda }+\eta^{\lambda }$[37], where $\tau^{\lambda }$ are the known continuous-spectrum expansion functions, $\eta^{\lambda }$ is an additive spatial white noise associated with the readout electronics dependent of each λ, and $\rho^{\lambda }$ is the raw signal.

The transmission spectrum can be written as the linear combination of known continuous-spectrum functions with unknown coefficients; based on simplicity and robustness of the method. In this case we use the Sum of Sine functions as the continuous-spectrum expansion functions for all the measured spectra. Thus, the generative model can be expressed as $\tau^{\lambda }\approx {\hat{\tau }}^{\lambda }=\sum_{n=1}^{N}a_{n}\sin\left(b_{n}\rho^{\lambda }+c_{n}\right)$. Here, $a_{n},b_{n}$, and $c_{n}$ are the unknown expansion coefficients that must be determined, and n=1,…,N are the number of coefficients of the function. The use of ${\hat{\tau }}^{\lambda }$ ensures an optimal approximation for $\tau^{\lambda }$.

The number of coefficients used to fit the curves are shown in Figure 4. The blue line represents one measured transmittance spectrum from the GS images cube. The segmented purple and black lines are the corresponding fit to *N* = 12 and *N* = 24 terms, respectively. As shown in Figure 4, *N* = 24 coefficients optimally fit all the measured spectra with a maximum root-mean-square error (RMSE) of less than 1%. Then, for each measured spectrum, the best fit is made using 24 Sum of Sines coefficients, which are considered the representation of the spectrum signal in this new space. It is fundamental to note that this allows us to make a significant dimensional reduction in each acquired spectra. Likewise, the curves will no longer be represented by high-resolution spectra but only by 24 expansion coefficients. This reduction is crucial for the processing cost in future real-time implementation.

Finally, to detect the presence of different types of syrups on two kinds of honey and classify them according to their level concentrations, we used three supervised learning algorithms: Linear discriminant analysis (LDA), Support vector machine (SVM), and Neural Network (NN). These classifiers presented better preliminary performance compared to other previously studied algorithms from the standard Matlab tool. (Machine learning tool, Matlab R2021b). Note also that the outputs of the classifiers are according to the purity ranges of the honey. A graphic summary of the proposed method is shown in Figure 5.

### 2.7. Dataset

For each sample, we obtained two hypercubes by using a transmittance microscopy imaging system. For each type of honey, we used 100 curves with 311 wavelengths, containing 5 levels of purity concentration. It is important to note that for a syrup concentration greater than 40%, the adulterated honey is visible to the naked eye. In this sense, the dataset focuses on syrups concentrations beyond this limit.

### 2.8. Performance Metrics

Five metrics were selected to evaluate the classification models’ performance: precision, recall, specificity, F1*scores*, and accuracy. The precision is the fraction of correct detections reported by the model, while the recall is the fraction of true events detected. At the same time, specificity is the ability to reject the samples of other classes [38,39]. In essence, precision, recall, and specificity are metrics that help us evaluate the predictive performance of a classification model on a particular class of interest and are calculated as follows:(1)$$Precision=\frac{TP}{TP+FP}$$
(2)$$Recall=\frac{TP}{TP+FN}$$
(3)$$Specificity=\frac{TN}{TN+FP}$$
where *TP* (true positive) is the samples correctly classified as belonging to a particular class, and *TN* (true negative) refers to the samples correctly assigned as not belonging to a specific class. *FP* (false positive) relates to the samples incorrectly classified as belonging and *FN* (false negative) refers to the samples assigned incorrectly as not belonging to a particular class [38]. On the other hand, accuracy is the rate of correct predictions to the total number of samples, so it is calculated by dividing the number of forecasts the model got right by the total number of predictions it made.
(4)$$Accuracy=\frac{TP+TN}{TP+TN+FP+FN}$$ By definition, F1*score* is the harmonic mean of precision and recall combined into a single number using the following mathematical expression:(5)$$F1scores=2\frac{Precision*Recall}{Precision+Recall}$$ Notice that F1*score* takes both precision and recall into account, which means it accounts for FPs and FNs. The higher the precision and recall, the higher the F1*score*.

## 3. Results and Discussion

### 3.1. Spectral Characteristics

Figure 6 shows an example of the transmittance spectra measured for the two kinds of honey and their respective levels of adulteration used in this work. Figure 6a–c show the cases of Q honey with the syrups F, G, and M, respectively. In addition, in Figure 6d–f are GS honey mixed with similar syrups in the previous case. Overall, we can observe the peculiarity of each syrups curve, which is easy to identify with the naked eye, and the degradation trend from pure honey to syrups, for each curve. To illustrate this, GS’s measured spectrums are noticeably degraded compared to adulterated GS honey (GS90, GS80, GS70, and GS60); this is for any syrup case (Figure 6d–f). This can be seen more clearly in the GS+F curves around the wavelength range between 450 and 630 [nm], where the pool of curves presents a more similar trend to the F curves. On the other hand, the case of Q honey is slightly different since it is possible to observe, concerning the intensity and forms, that the degradation of the curves overlaps each other, being difficult to distinguish with the naked eye, as shown in Figure 6a,b.

### 3.2. Training and Model Validation

For this analysis, we trained and validated the model by selecting 10-fold cross-validation to protect the overfitting. We built the feature matrix using 100 spectral samples of the level of honey purity obtained from the hyperspectral cube of each kind of honey. From each spectral sample, 24 coefficients were extracted using the regression technique as indicated in Section 2.6. Then the classification process for each purity range of honey was performed using an LDA model with a complete covariance structure, a quadratic SVM model with an automatic kernel, and a feedforward NN model with a fully connected layers size of 100 and a rectified linear unit (ReLU) activation function.

The values of metrics obtained by the classifiers for each class are summarized in Table 2 and Figure 7. Firstly, in the case of Q honey, specifically Q + F, the LDA classifier achieved the best performance with an average of 93.4, 93.2, and 97.6% in precision, recall, and specificity, respectively; however, when analyzing the values of F1 scores in Figure 7a, it can be observed that the highest scores (98 and 98.4%) in the Q90 and Q70 classes were obtained by the SVM classifier. This is because the balance between recall and precision values obtained by SVM are higher than LDA values in the Q90 and Q70 classes. On the other hand, note that the LDA score values for the Q80 class far outweigh the different classifiers, caused by the 91.9% correct predictions of samples compared to the 82 and 78% of SVM and NN. Besides, in the Q + M combination, SVM shows the best averages with 89.2, 89.3, and 96.3% above the other classifiers. A similar occurrence happens for the scores of Figure 7c, with 91, 83, and 83% for the Q90, Q80, and Q70 classes, respectively, observing that the score in Q60 class is equal to 100% to SVM and LDA. Note that NN has the lowest results overall; however, the specificity average (94.5%) slightly exceeds that of LDA (94.3%) in type Q + M. Finally, in the Q + G mixture, all classifiers obtained perfect results (100%).

Secondly, the results obtained in GS honey show that to GS + F, the LDA classifier achieved the slightly best performance with precision, recall, and specificity averages of 96.2, 96.2, and 98.6%, compared to the 95.2, 95.2, 98.3, 93.4, 91.5, and 97.8% of SVM and NN, respectively. Again, NN shows lower averages of 93.4, 91.5, and 97.8% than the other classifiers; however, as shown in Figure 7d, it is noteworthy that NN achieved higher and second-best results for GS90 and GS70 classes, respectively. In the case of GS + G, SVM shows superiority with 95.9, 96, and 98.7% in the respective average values for each metric. On the other hand, in Figure 7e, it is observed that NN overtook SVM with results of 93 and 91.8 in the GS70 class and obtained the second-best mark in the GS90 class. Finally, GS + M, LDA, SVM, and NN obtained the best result averages in different metrics. LDA achieved a 95.1% in specificity, SVM an 85% in the recall, and NN a 94.7% in precision. Focusing on Figure 7e, LDA achieved the best result with an 84% for GS80 class. Meanwhile, SVM shows the best performance with 85.6 and 77% in GS90 and GS70 classes. Overall, we note that the GS+M combination recorded the lowest values by category.

Finally, from Figure 8a–c, the accuracies of the classifiers for the different cases of honey mixed with syrups are shown. For the case of the LDA classifier shown in Figure 8a, the accuracy values are 93.2, 100, 85, 96.2, 93.2, and 84.2% for the classes Q + F, Q + G, Q + M, GS + F, GS + G and GS + M, respectively. In Figure 8b, for the SVM classifier, they are 91.2, 100, 89, 95.2, 96, and 85%, and in Figure 8c, for the NN classifier, the values are 87.8, 100, 83.8, 93.5, 95.2, and 84.5%, all for the same classes of mixtures shown in Figure 8a. At first glance, LDA reached good values for Q + F and GS + F classes; however, for GS + G and GS + M classes, LDA obtained the lowest value of all the classifiers. Additionally, NN underperformed overall, only achieving the second-best accuracy in the GS + G and GS + M cases. Lastly, SVM showed the best performance for all classes, highlighting its performance in Q + M, GS + G, and GS + M classes.

In summary, It should be noted that when the two types of honey were mixed with G syrup, the classifiers performed the best. Specifically for Q + G, as seen in Figure 6c, the curves are superimposed on each other, which makes it challenging to make a classification using these curve pools. However, all the classifiers’ performances were perfect, demonstrating the feature extraction method’s robustness. This can be ratified by looking at F1 scores and accuracy in Figure 7b and Figure 8. On the contrary, the lowest values were for the honey types mixed with syrup M, particularly for the GS70 class, as shown in Figure 7f and Figure 8. For these cases, the values determined by the coefficients function have very similar terms, making the task more difficult for classifiers. As a possible solution to this problem, it would be good to try feature extraction by focusing on where the difference is more significant between the pool of curves.

Our results are justified not only in the context of endemic honey characterization but also in the benefit of hyperspectral microscopy. A very recent work published by Zhang et al. in 2022 [40] compares different classifiers performance to categorize honey samples from New Zealand. In that work, botanical origin was categorized using four different machine learning algorithms, showing that SVM achieved >99% in accuracy rate, which is in agreement with our results. In addition, in Ref. [41], three different machine learning algorithms were used to predict honey floral origin using spectral information. Additionally, advantages have been discussed in relation to snapshot microscopy and spectrometry [42,43]. On one hand, there is no more sophisticated type of XY scanning of samples or method of limiting the exposure time of remote sensing acquisitions. On the other hand, the use of a microscopic field of view allows spectral analysis of each pixel and, with Snapshot instruments, SNR tends to enhance.

## 4. Conclusions

The spectral signature, and the acquired data presented in this work, allowed us to classify three types of syrups in two valuable endemic types of honey from Chile; Guindo Santo and Quillay. By using classification algorithms based on artificial learning, it was possible to determine if the honey had been adulterated at concentrations below the human recognition capabilities. The estimation reached an error under 1% (RMSE equal to 0.0037 for both honeys) using 24 coefficients in the function fit. In addition, the number of curves to train was reduced, from 106,926 to 100, which are selected randomly. Therefore, a feature matrix of the size (100 × 24), corresponding to 100 normalized transmittance spectral curves and the 24 estimation coefficients, was achieved for the classification algorithm. This reduction strategy is significant so as to lower the time processing cost in future real-time applications. The best precision was obtained by SVM, followed by LDA and NN classifiers. In future work, new functions will be explored by generating estimation curves with greater accuracy. Additionally, new hyper-parameters of the SVM will be investigated to improve its performance in order to increase average validation precision above 95% for each sample and all adulterating factors.

The hyperspectral analysis performed on the Chilean native and endemic honey, Quillay and Guindo Santo, allows for the determination of their spectral signature with high reliability. These results, and the fact that these types of honey are a natural food unique in the world in terms of their origin and production, allow projecting its use by Chilean beekeepers to optimize their honey production business. Certification of these types of honey facilitates their commercialization on national and international markets, ensuring fair economic returns to the communities of producers. Additionally, certification prevents the sale of adulterated products that lack adequate indications of their floral origin and provenance, stating that they are Chilean Quillay and Guindo Santo honey, but which they are in fact not.

## Figures and Tables

**Figure 1 foods-11-03868-f001:**
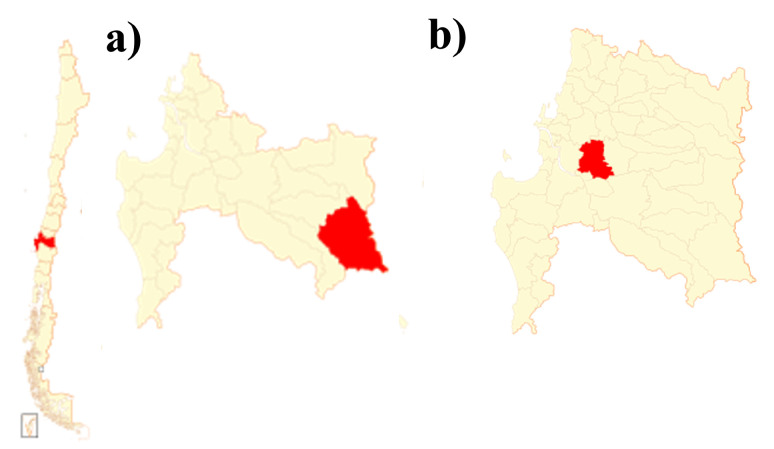
Map of Chile highlighting (**a**) the Pitril sector, Alto Biobío Commune, and (**b**) Yumbel. Both areas are located in the Biobío Region between 36°26′ and 38°29′ south latitude.

**Figure 2 foods-11-03868-f002:**
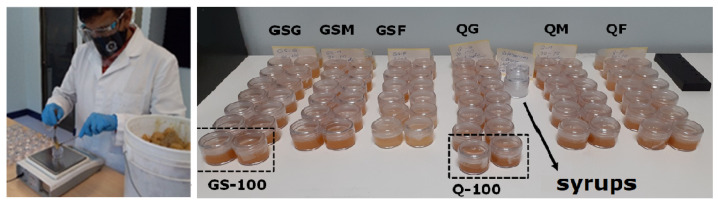
Sample preparation used for the acquisition of the transmission spectrum of honey from Guindo Santo and Quillay mixed with different syrups (Fructose, Glucose, High Maltose). View of the mixing preparation and all samples used. GSG, GSM and GSF are indicated for Guindo Santo honey mixed with Glucose, High Maltose, and Fructose syrups, respectively, and QG, QM, and QF for Quillay honey mixed with the same syrups. The references correspond to two samples of 100% honey for each case (GS-100 1 and GS-100 2 for Guindo Santo and Q-100 1 and Q-100 2 for Quillay) and 100% syrups, one for each case (G-100: 100% Glucose; M-100: 100% High Maltose; F-100: 100% Fructose).

**Figure 3 foods-11-03868-f003:**
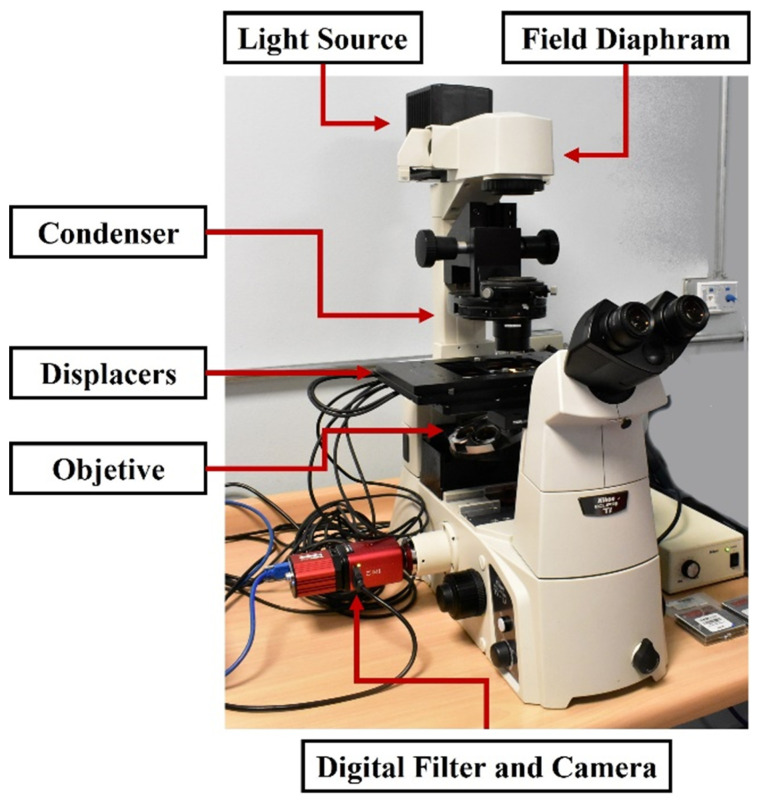
Hyperspectral microscope system. The setup also includes a high-speed motorized XY scanning stage (MLS203 Base, Thorlabs) with a DC motor controller (BBD302, Thorlabs) for sample positioning. The whole control and measurement system is programmed in Labview as a graphical user interface.

**Figure 4 foods-11-03868-f004:**
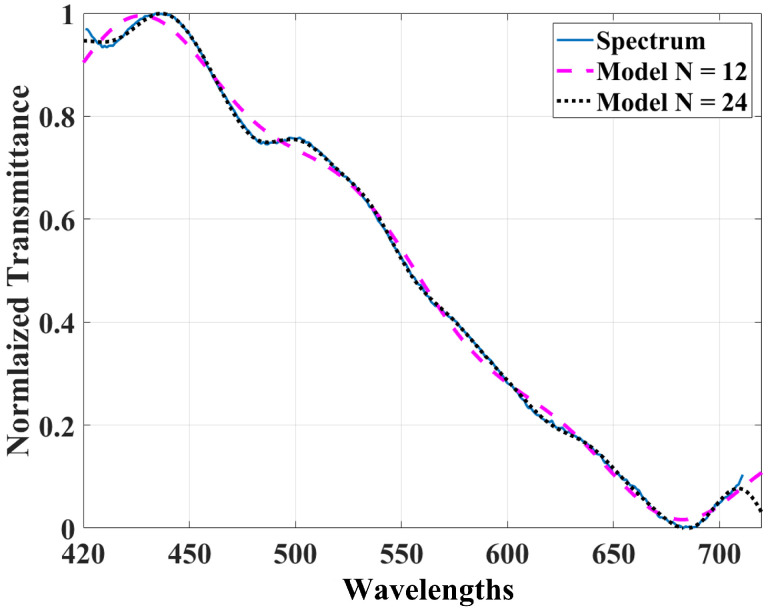
Adjust the measured transmittance spectrum from the GS images cube using the Sum of Sine functions. The segmented purple and black lines represent a fit using 15 and 24 coefficients that obtained an RMS of 0.016333 and 0.003743, respectively.

**Figure 5 foods-11-03868-f005:**
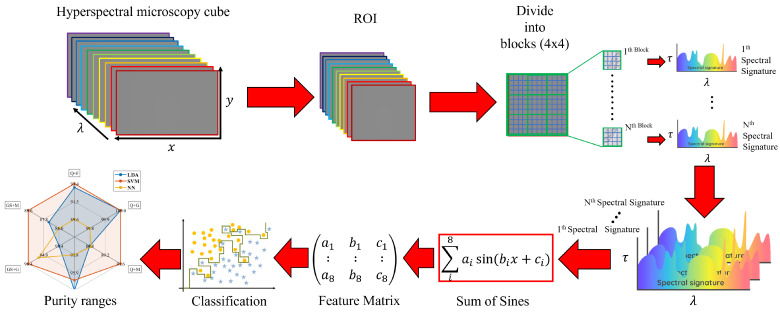
Block diagram of the adulterated honey detection algorithm. Firstly, the transmittance is calculated from each pixel standing in the hypercube’s region of interest (ROI). Then, the ROI is subdivided into 4 × 4 neighborhoods and averaged to create the spectral curves. Secondly, we used the Sum of Sine functions to generate regression coefficients of all the measured spectra. Twenty-four coefficient features were extracted from each spectral curve, which are introduced in the classification section. Thirdly, and finally, we used three classifiers to detect the presence of different syrups on the two kinds of honey and classify them according to their concentrations.

**Figure 6 foods-11-03868-f006:**
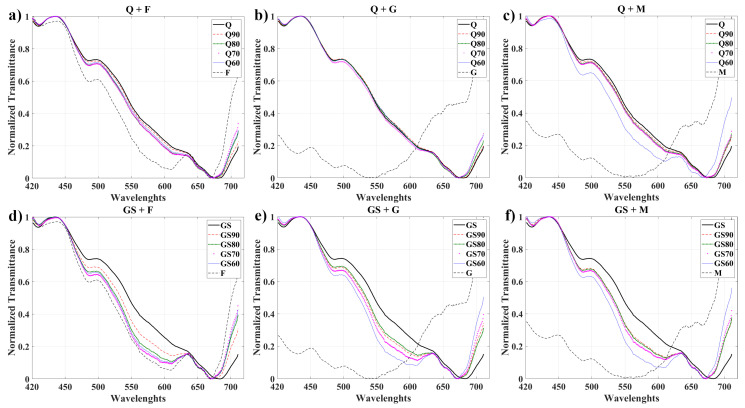
The transmittance curves were obtained from hyperspectral microscopy imaging. From (**a**–**f**) are the Q and GS honey curves with different concentration levels using F, G, and M syrups, respectively.

**Figure 7 foods-11-03868-f007:**
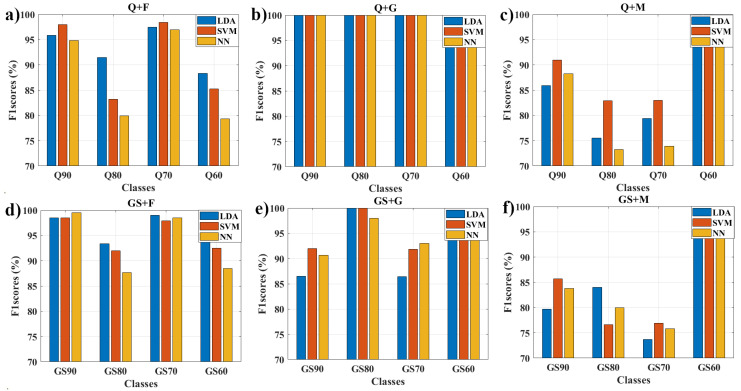
Bar plots for F1 scores rate (validation) of Table 2, 10-fold cross-validation results LDA, SVM, and NN classifiers using 24 coefficients. From (**a**–**f**) are the Q and GS honey scores with different concentration levels using F, G, and M syrups, respectively.

**Figure 8 foods-11-03868-f008:**
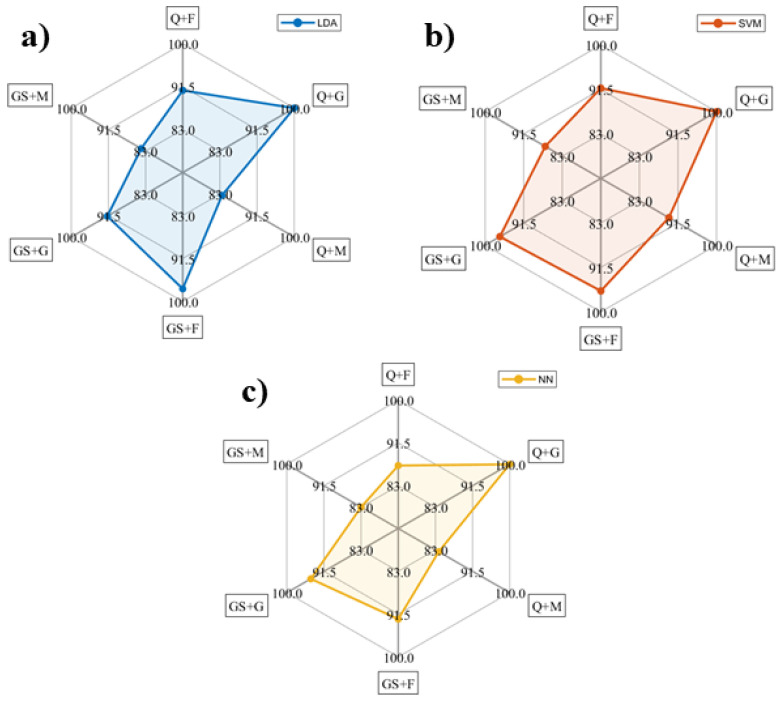
Training results. Comparison of the accuracy (validation) rate achieved by the different classifiers for each combination between Q and GS honey and F, G, M syrups. The results of classifiers LDA, SVM, and NN are represented in (**a**–**c**), respectively.

**Table 1 foods-11-03868-t001:** Values of percentages and masses of the mixtures analyzed correspond to Quillay and Guindo Santo’s honey samples separately.

Honey/Total Mass (%)	Honey Quantity (g)	Syrup Quantity (g)
100	23	0
90	20.7	2.3
80	18.4	4.6
70	16.1	6.9
60	13.8	9.2

**Table 2 foods-11-03868-t002:** The Precision (P), Recall (Re), and Specificity (Sp) metrics obtained for 10-fold cross-validation in each class are shown.

Class		Classifier									Class		Classifier								
		LDA			SVM			NN					LDA			SVM			NN		
		P	Re	Sp	P	Re	Sp	P	Re	Sp			P	Re	Sp	P	Re	Sp	P	Re	Sp
**+F**	**Q90**	98.9	93	99.6	98	98	99	96.8	93	99	**+F**	**GS90**	97	100	99	98	99	99.3	99	100	99
	**Q80**	91.9	91	97.3	84.5	82	95	78	82	92		**GS80**	94.8	92	98	92	92	97.3	97	80	96
	**Q70**	97.9	97	99	98.9	98	99.6	97	97	99		**GS70**	100	98	100	98.9	97	99.6	100	97	100
	**Q60**	85	92	94.6	83.6	87	94.3	79.7	79	93		**GS60**	93	95	97.6	92	93	97	88	89	96
**Average**		**93.4**	**93.2**	**97.6**	**91.3**	**91.2**	**97**	**87.8**	**87.8**	**95.7**			**96.2**	**96.2**	**98.6**	**95.2**	**95.2**	**98.3**	**93.4**	**91.5**	**97.8**
**+G**	**Q90**	100	100	100	100	100	100	100	100	100	**+G**	**GS90**	86	87	95	90	94	96.6	88.5	93	96
	**Q80**	100	100	100	100	100	100	100	100	100		**GS80**	100	100	100	100	100	100	100	96	100
	**Q70**	100	100	100	100	100	100	100	100	100		**GS70**	86.8	86	95.6	93.7	90	98	93	93	97.6
	**Q60**	100	100	100	100	100	100	100	100	100		**GS60**	100	100	100	100	100	100	99.7	99	100
**Average**		**100**	**100**	**100**	**100**	**100**	**100**	**100**	**100**	**100**			**93.2**	**93.2**	**97.6**	**95.9**	**96**	**98.7**	**95.4**	**95**	**98.4**
**+M**	**Q90**	89	83	93.6	91	91	97	89.6	87	96.6	**+M**	**GS90**	75	85	92	79.4	93	92	84.6	83	94.6
	**Q80**	75	76	91.6	80.9	85	93	69	78	88.6		**GS80**	84	84	94.6	81.8	72	94.6	80	80	93
	**Q70**	77	82	92	85	81	95	77	71	93		**GS70**	79	69	94	78.9	75	93	74.7	77	91
	**Q60**	100	100	100	100	100	100	100	99	100		**GS60**	100	100	100	100	100	100	100	98	100
**Average**		**84.2**	**85.2**	**94.3**	**89.2**	**89.3**	**96.3**	**83.9**	**83.7**	**94.5**			**84.5**	**84.5**	**95.1**	**85**	**85**	**94.9**	**94.7**	**84.5**	**94.6**

## Data Availability

Data is contained within the article.

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
