# Peer review of "Hyperspectral Microscopy Technology to Detect Syrups Adulteration of Endemic Guindo Santo and Quillay Honey Using Machine-Learning Tools"

_foods, 2022, doi:10.3390/foods11233868_

Round 1

Reviewer 1 Report

Summary: This paper reports a system combined with intelligent algorithm for syrups adulteration of endemic honey classification. Three classifiers were used to detect the presence of different types of syrups on two kinds of honey and classify them according to their level concentrations. Especially, the authors claim SVM is better than LDA and NN in average accuracy.

Overall comment: The manuscript is well-written with the clear structural framework. However, the weaknesses of the manuscript include a number of details that are missing and errors throughout the manuscript. Below are some of the mistakes and inconsistencies that I found.

Q1. Line 74-75. “With the solution present in this work, ……Guindo Santo honey would have an immediate competitive advantage ……” This sentence is obviously overstated. The sample preparation and detection techniques in this paper are carried out under laboratory conditions, and whether they can be widely applied in practice remains to be considered.

Q2. Line 91-92. How to ensure “when the species under study began producing nectar, the bees produced only Guindo Santo and Quillay honey.” Please provide more details in honey sample collection.

Q3. Line 93. It seems that honey samples are extracted directly. Please provide more details about the acquisition and preservation of honey samples.

Q4. Line 110-115. What is the purpose of “The melissopalynological analysis”? Please clarify it.

Q5. Figure 4, what is RMS? What is Wavelengthts? Check the spelling.

Q6. Figure4 and Line 177-186, Why N=24 is the best? It seems the author want to reduce the dimension of the wavelengths, but why not use spectral extraction method or reduction method?

Q7. Line 236-237. How to determine the error in accuracy rate is caused by similar terms in the training coefficient?

Q8. Line 246. It is confusing, according to which score, NN is the second-best accuracy? 

Reviewer 2 Report

In the present paper authors used hyperspectral microscopy technique in conjunction with chemometric techniques to detect the honey adulteration with fructose, glucose and maltose. It is quite interesting work but the quality of presentation should be increased. The authors should address the following issues in the manuscript

1.     Abstract part should include Introduction (2-3 lines), aim of the work (1-2 lines), methodology (3-4 lines), results (3-4 lines) and conclusion (2 lines). Rewrite the abstract.

2.     Line 57-80 it is entirely materials and methods it doesn’t require here. Please refer introduction section of other research papers.

3.     In Table 1 honey quantity and syrup quantity values separated with, is it “.”? Cross check it.

4.     Line 188-194 authors should discuss about the total of number of samples, classification of data for model development and validation and conditions adopted during development of classification models such number of neurons, hidden layers, transforming function similarly for SVM and LDA also.

5.     The author classification measures such as sensitivity, specificity etc. authors take help from the following paper

https://doi.org/10.1016/j.compag.2020.105539

6.     Line 201-213 move this part to results section and create a separate section “Spectral characteristics”.

7.     The entire results and discussion section completed with classification accuracy and the same information was presented in both in tabular and pictorial form, remove either of these.

8.     The results and discussion section is not clear and discussion part is very poor. The obtained results should be justified with previous studies.

9.     The discussion part should be strengthen with overall outcome, recommendations and future scope.

Round 2

Reviewer 2 Report

Discuss about classification measures such as precision, specificity, and recall in the materials and methods section also. Authors should take the help of English native speakers, still, there are some grammatical corrections are required before accepting the paper. 

Author Response

We thank the reviewer for the suggestion. The discussion about metrics was moved to the materials and methods section, and we have also corrected the text's error grammar.